# Comparing End-of-Life Vehicle (ELV) and Packaging-Based Recyclates as Components in Polypropylene-Based Compounds for Automotive Applications

**DOI:** 10.3390/polym16131927

**Published:** 2024-07-06

**Authors:** Markus Gall, Daniela Mileva, Wolfgang Stockreiter, Christophe Salles, Markus Gahleitner

**Affiliations:** 1Borealis Polyolefine GmbH, Innovation Headquarters, St Peterstr. 25, 4021 Linz, Austria; markus.gall@borealisgroup.com (M.G.); daniela.mileva@borealisgroup.com (D.M.); wolfgang.stockreiter@borealisgroup.com (W.S.); 2Borealis Services S.A.S., 12 rue de Londres, 75015 Paris, France; christophe.salles@borealisgroup.com

**Keywords:** polypropylene, recycling, automotive, morphology, mechanics

## Abstract

Increasing recycled plastic content in cars to 25% by 2030 is one of the key measures for decarbonizing the automotive industry defined by the European Commission. This should include the recovery of plastics from end-of-life vehicles (ELVs), but such materials are hardly used in compounds today. To close the knowledge gap, two ELV recyclate grades largely based on bumper recycling were analyzed in comparison to a packaging-based post-consumer recyclate (PCR). The composition data were used to design polypropylene (PP) compounds for automotive applications with virgin base material and mineral reinforcement, which were characterized in relation to a commercial virgin-based compound. A compound with a 40 wt.-% ELV-based bumper recyclate can exceed one with just a 25 wt.-% packaging-based recyclate in terms of stiffness/impact balance. While the virgin reference can nearly be matched regarding mechanics, the flowability is not reached by any of the PCR compounds, making further development work necessary.

## 1. Introduction

Polypropylene (PP) is one of the thermoplastic polymers with the biggest global production volumes, reaching about 75 million tons (Mt) in 2020, with a clear trend towards circularity [1]. At the same time, even in Europe, only 35% of all polymer waste is being recycled, and according to Plastics Europe, only about 9% of all thermoplastic polymers used are based on recycled materials [2]. The proportion of mechanical recycling is at 45% higher for materials based on the collection of packaging materials, limiting the performance range of the thus-produced recyclates. While steel and aluminum recycling are pretty much standard elements of end-of-life vehicle (ELV) recovery, the contribution of these processes to plastics recycling is very limited.

The European Commission has defined a target for recycled plastic content in cars of 25% by 2030, of which 25% should come from closed-loop ELV treatment [3]. This is only one of a package of measures defined in the cited proposal aiming at an annual reduction of 12.3 million tons of carbon dioxide equivalents (CO2-eqs.) in 2035, which is a key contribution to decarbonizing the automotive industry. The measures can also be seen as one of the elements of the “ReShaping Plastics” program proposed by Systemiq [4] to achieve plastic circularity of 78% in Europe by 2050. Due to global trade connections, similar targets can be expected for other world regions as well in the future.

The role of PP-based materials is essential here due to their relatively large share of the overall plastic content of contemporary vehicles of about 30–40%, according to Matos et al. [5]. This is higher than in other application areas, and even though the usage period is longer in the automotive sector, the proportions of all polymers that have been recycled since their first market introduction in the 1960s are generally low. According to Geyer et al. [6], by 2015, only about 9% of all produced polymers had been recycled, while 12% were incinerated and 79%—more than 6 billion tons—ended up in landfills or the environment. More recent and detailed figures in this respect can be found in material flow analyses recently performed in Austria [7] and Europe [8]. The Austrian study focuses on application segments, highlighting the positive trend in plastic waste treatment between 1994 and 2010. In that period, the effective elimination of landfilling was achieved mostly by moving to incineration, while the mechanically recycled fraction still remained at 9%. In contrast, the European study covers the years 1950 to 2016 and presents a polymer-specific analysis. The fate of the 310 Mt of PP consumed in Europe in that period is given as follows: 48% landfilled, 16.5% incinerated (including the use as fuel substituent in the steel and cement industries), 13.7% “in-use stock” (including automotive usage), and 7.5% recycled or re-used. The balance is material exported outside Europe or that has an unclear fate. The recycling fraction for PP given in that study is clearly lower than that for poly(ethylene terephthalate) (PET) or any of the two polyethylene (PE) classes, high-density PE (HDPE) and low-density PE (LDPE). This analysis is certainly more comprehensive and far more detailed than the one used for the aforementioned Systemiq study [4].

Less data on polymer recycling in general are available for other world regions. A recent study for the US gives an overall recycling rate of just 6.2%, with 77% of all plastic waste going to landfill sites [9]; for China, the nation with the biggest polymer production worldwide, an average recycling rate of 29% for the period 1978–2019 is claimed [10], with 42% of polymers being landfilled and 17% discarded without any proper treatment. No specific data for ELV recycling are available in either case.

Another relevant angle for ELV recycling is the life cycle analyses (LCAs) of cars and the passenger transportation system in total, which will only be discussed regarding relevant aspects here. While the use phase of a vehicle with a combustion engine is generally responsible for the largest environmental impact of its life cycle, both construction and disposal must also be considered. In a Dutch study from 2003 [11], the increased use of polymer components for reducing the weight of passenger cars is highlighted, together with the fact that only a small fraction (<10%) of this ends up in disassembled parts, while the majority (~90%) becomes automotive shredder residue (ASR) as a side stream of steel recycling. A more recent Belgian study from 2016 [12] already considers an earlier version of the EC proposal, treating the polymer-related ASR fraction collectively as ‘shredder light fraction’ (SLF) and giving it little relevance in the overall LCA. The SLF is a collective light material fraction retrieved after milling the car body and, thus, comprises not only bumper elements but also interior and under-the-hood elements.

More relevant for the present study are papers that focus on LCAs of plastic components in various applications, mostly dealing with the recycling of short-lived products like packaging, but also considering components of electrical equipment and vehicles. The biggest potential for carbon footprint reduction is estimated for mechanical recycling [13], with the collective treatment of bigger material streams showing further advantages [14]. Few papers specifically deal with LCAs of plastics in the automotive sector: A study initiated by Renault Group from 2019 analyses the potential of closed-loop polypropylene recycling [15], concluding that increased PP recovery can generate a volume of 52 kt of recycled plastic for use in vehicle production. A more general screening of the automotive sector by Matos et al. [5] sees great potential for circularity, but also severe limitations resulting from design complexity and multi-polymer usage. An agreement with the more general LCA studies is that integrating different waste streams, like ELV and electronics recycling, may improve the situation.

As for the LCA studies, studies on practical aspects of mechanical recycling are mostly related to packaging or, more generally, to post-consumer plastic waste. Reviews in the field [16,17,18] consequently consider differentiated waste collection as the main source of post-consumer recyclates (PCRs), the composition of which reflects typical polymer selection for packaging applications. Polyolefins, i.e., the different types of PE and PP, dominate the composition next to PET, with a minority fraction of polystyrene (PS) and a significant sorting residue comprising other polymers and multilayer constructions [19,20,21]. While the latter are without doubt appropriate solutions for fulfilling complex requirements of mechanics and barrier, the respective polymer combinations like PET/PE with ethyl–vinyl alcohol copolymer (EVOH) as an oxygen barrier make mechanical recycling difficult to impossible due to the resulting multiphase systems with poor compatibility [22].

Cross-contamination effects between PP and PE [23], and especially between polyolefins and non-polyolefins [24], reduce the performance of PCR materials in cases of poor sorting quality. Phase structure, interfacial adhesion, and the relationship in mechanics between the phases are decisive factors, similar to multiphase polymer design in general. This calls for a close analysis of such recyclates prior to their application in designing compounds [25] and makes the use of specific compatibilizers and modifiers for improving phase adhesion necessary [26]. The latter is less of an issue in automotive compound design, as PP impact copolymers and PE-based plastomers are commonly used as components anyway.

Compared to the vast literature on packaging recycling, specific papers dealing with ELV recycling are rather rare. Probably, the earliest example of using ASR in part design is the study of Robson and Goodhead from 2003 [27], who applied a skin-core design with virgin material on the part surface and an ASR compound in the core. In up to five cycles, a massive drop in tensile strength and energy to break was found when compared to virgin PP. From the same time is the bumper recycling study of Luda et al. [28]. They put the focus on the degradation effects of PP-based bumper compositions in mechanical recycling, finding clearly positive effects of both paint stripping and re-stabilization prior to compounding.

Rabeau Epsztein et al. (2014) [29] compared the performances of three samples of recycled polyolefins from ELVs to be used in automobile applications, specifically as matrices for short glass fiber (SGF)-reinforced compounds. The PE content in the tested recyclates was rather high, limiting the stiffness and heat resistance of the final products. The work of Kozderka et al. [30] is rather a model study on the recycling of pure high-impact polypropylene (HIPP), the most common polymer in car body parts such as bumpers. HIPP mechanical properties were found to deteriorate from the first reprocessing step, albeit without significant effects before the sixth reprocessing step.

Yang et al. [31] used a rather special approach for incorporating ASR into compounds, applying solid-state shear milling (SSSM) and a mixing ratio of 70/30 to 10/90 (by mass) for the production of recycled PP/ASR compositions. They think that it is an advantage to avoid complete melting and found an acceptable performance of up to 50 wt.-% ASR content, increased further by adding a compatibilizer. Maleic anhydride-grafted styrene elastomer (SEBS-g-MAH) was found to give the best toughness. A very specific study related to the essential step of paint removal and its effects on PP bumper recyclate performance was carried out by Guo et al. [14]. Efficient paint stripping was especially found to improve the ductility of the recycled material significantly.

Recently, there have also been ideas about alternative fabrication techniques for automotive components like 3D printing [32]. While this technology is not likely to substitute high-speed conversion processes like injection molding for bigger parts of a car like bumpers, it has already been explored regarding its suitability for applying recycled polymers [33]. It is, however, difficult here as well to manage the balance between the flowability, solidification speed, and mechanical performance.

The present study had the following dual target:Analyzing semicommercially (i.e., as samples from pilot scale) available ELV recyclate grades based on bumper or ‘shredder light fraction’ (SLF) recycling in terms of composition and suitability for automotive compounds;Testing these as components in automotive compounds in direct comparison to a packaging-based post-consumer recyclate (PCR) already established as a compounding component.

The performance targets for the final compounds were based on typical specifications of original equipment manufacturers (OEMs), i.e., car brand owners, for passenger vehicle bumpers. For this step, the performance was further compared to a virgin-based commercial compound.

## 2. Materials and Methods

### 2.1. Investigated Recyclates and Modifiers

Shredded flakes from bumper recycling in Germany (PCR-ELV-bumper) and the polymer shredder light fraction (SLF) in the Netherlands (PCR-ELV light fraction) were applied, both being produced in semicommercial scale and received under obligation of not disclosing the precise source. Prior to further characterization and compounding, the irregular-shaped flakes were pelletized on a twin-screw extruder ZK50 (Collin GmbH, Maitenbeth, Germany) with an L/D-ratio of 22, using a 200 µm screen-changer-type melt filter (same supplier), followed by strand pelletization. This was carried out not only to homogenize the material but also to remove impurities like paint and metal residues. In this compounding step, 0.30 wt% of a 1:1 mixture of pentaerythrityl-tetrakis(3-(3′,5′-di-tert. butyl-4-hydroxyphenyl)-propionate (Irganox 1010 from BASF SE, Ludwigshafen, Germany) and Tris (2,4-di-t-butylphenyl) phosphite (Irgafos 168 from BASF SE, Ludwigshafen, Germany) were added to re-stabilize the material. The third recyclate was a PCR-Packaging material based on Borealis’ own pilot-scale operations. Table 1 summarizes the key analytical parameters and composition data for all PCRs (see Section 2 for details), the differences of which will be discussed in Section 3.

The optical appearances of the flakes and the as-extruded pellets for the PCR-Packaging and PCR-ELV-Bumper materials are shown in Figure 1, highlighting the effect of homogenization.

One of the two PCR-ELV materials, the SLF type, was found to be too soft and too rich in PE, limiting its stiffness and heat resistance. Compounds were, therefore, only prepared based on two of the materials from Table 1. Depending on the PCR composition, different compositions of virgin base copolymers [34,35] with the addition of a PE-plastomer [36,37] were selected.

The melt flow rate (MFR, 230 °C/2.16 kg) and composition data for these two ‘virgin mixes’ are listed in Table 2. Due to the high mineral residue content in the PCR-ELV-Bumper, which was identified to be mostly talc via X-ray fluorescence spectroscopy (Mg, Al and Si as dominant elements), the reinforcing filler content was also adapted. As a reinforcing mineral filler, talc Steamic 1CA of Imerys, France, with a median diameter d_50_ of 1.8 µm and top cut diameter d95 of 6.2 µm, was used.

A commercial virgin-based compound of Borealis AG (Vienna, Austria) was used as a reference material in terms of performance. It is characterized by an MFR of 22 g/10 min and a mineral filler content of 15 wt.-%.

### 2.2. Analytics and Mechanics

Standard characterization methods, as also used before for recyclates and their compositions [25], were used. MFR was measured according to ISO 1133 at 230 °C and a 2.16 kg load [38]. Differential scanning calorimetry (DSC) was run according to ISO 11357/part 3/method C2 [39] in a heat/cool/heat cycle with a scan rate of 10 °C/min in the temperature range of −30–26 to +225 °C with a TA Instrument Q200 performing differential scanning calorimetry (DSC) on 5 to 7 mg samples. The two melting temperatures corresponding to the PE (T_m,PE_) and PP content (T_m,PP_) and the two corresponding melting enthalpies (H_m,PE_ and H_m,PP_) were determined from the second heating step. Density was measured according to ISO 1183-1/2004 method A [40] on compression-molded specimens. Thermogravimetry (TGA) was used to determine the inorganic content of the PCR materials according to ISO 1172:1996 [41] with a Perkin Elmer (Waltham, MA, USA) TGA 8000. Approximately 10–20 mg of material was placed in a platinum pan, and temperature was equilibrated at 50 °C for 10 min and afterwards raised to 950 °C under nitrogen at a heating rate of 20 °C/min. The ash content was evaluated as the wt.-% at 850 °C.

For determining the crystalline (CF) and soluble (SF) fractions and their respective properties (intrinsic viscosity, IV, and ethylene content, C2), the CRYSTEX method [42] was applied. A CRYSTEX QC apparatus of PolymerChar Valencia, Spain, was used for this analysis, which has developed into an excellent alternative to the frequently used separation of PP copolymers and compositions into xylene cold soluble and insoluble fractions (XCS/XCI), followed by analysis of the fractions. The crystalline and amorphous fractions are separated through temperature cycles of dissolution at 160 °C, crystallization at 40 °C,0 and re-dissolution in 1,2,4-trichlorobenzene at 160 °C. The quantification of SF, CF, and ethylene content (C2) was achieved by means of an integrated infrared detector (IR4), and for the determination of the intrinsic viscosity (iV), an online 2-capillary viscometer was used.

All mechanical tests were carried out on injection-molded specimens prepared in line with ISO 19069-2 [43] on an Engel e-motion 310/55 machine (Enel Austria GmbH, Schwertberg, Austria) using a melt temperature of 230 °C for all materials, irrespective of the material melt flow rate. Flexural modulus was determined according to ISO 178/method A [44], and Charpy notched impact strength (NIS) was determined according to ISO 179 1eA [45], on test bars of 80 × 10 × 4 mm^3^. Instrumented puncture tests (IPT) were conducted in accordance with ISO 6603-2:2023 [46] on plaques of 60 × 60 × 3 mm^3^. The coefficient of linear thermal expansion (CLTE) was determined in accordance with ISO 11359-2 [47] on 10 mm long pieces cut from the centers of same injection-molded specimens used for the flexural modulus determination. The dimension of the CLTE specimen was 10 × 10 × 4 mm^3^, testing only in machine direction (MD) for the present study. The measurement was performed in a temperature range from −30 resp. +23 °C to +80 °C at a heating rate of 1 °C/min.

The morphology of compounds was evaluated via scanning electron microscopy (SEM) on cryo-cut surfaces after staining with RuO_4_ [48] on injection-molded specimens as for the flexural test. An Apreo S LoVac microscope of ThermoFisher Scientific (Waltham, MA, USA) was used.

### 2.3. Compounding

Compounds targeted at property profiles typical for automotive exterior components like bumpers or side trims were prepared based on two of the recyclates. A ThermoFisher (USA) TSE24 twin-screw extruder with an L/D ratio of 40, a high-intensity mixing screw configuration, and a temperature profile between 190 and 220 °C was used, followed by melt strand cooling in a water bath and strand pelletization. The following further additives were used in compounding: 0.20 wt.-% of the slip agent oleamide and 0.05 wt.-% of the antioxidant octadecyl 3-(3′,5′-di-tert. butyl-4-hydroxyphenyl)propionate. Both additives were dosed in a blend with powder of polypropylene homopolymer for better dispersion in the final compound, with the amount of said blend not exceeding 1 wt.-% of the total composition. Compound compositions and all analytical and mechanical characterization results are presented in Table 3.

## 3. Results

### 3.1. ELV and Packaging Recyclates

The compositions of the three PCR materials differ significantly, reflected by their analytical and mechanical properties in Table 1. Based on a combination of DSC (see Figure 2) and CRYSTEX analysis, one can calculate the approximate fractions of PP, PE and amorphous ethylene–propylene copolymer (EPC) in the respective types. For finding the crystalline PP and PE contents, the H_m_-values related to the two polymers were normalized to the respective enthalpies for fully crystalline polymers, 293 J/g for PE [49] and 170 J/g for PP [50]. The result of this calculation, which assumes an equal degree of crystallinity of both polymers in the composition, was then related to the crystalline fraction (CF) from CRYSTEX, assuming SF to be identical to the EPC content in line with earlier work [39].

A rough idea of the relationship between crystalline PP and PE can already be obtained from the heating scans in Figure 2, including the fact that T_m_ also varies for both parts. Two of the three PCRs have a certain EP-random copolymer content, as indicated by the lower T_m,PP_, and in none of the compositions is the PE really HDPE, which would have a T_m,PE_ of >130 °C. The diagram in Figure 3 presents the full results of said calculation, already excluding the filler fraction, which is high only for the PCR-ELV-Bumper type (18 wt.-%). Both ELV-PCRs show higher EPC contents, but the ‘light fraction’ (SLF) material clearly sticks out in terms of PE content. While the other two PCRs bear at least some compositional resemblance to a virgin PP impact copolymer [51,52], this one is instead a PP/PE blend with the well-known limitations of such compositions [23,25].

Three further important factors for the significantly different mechanical performances need to be considered:The molecular weight of elastomer-like EPC, as well as its composition and content, is decisive for the stiffness-impact balance of impact copolymers and similar compositions [35,51]. The PCR-Packaging type clearly has the lowest C2(SF), related to a rather high glass transition temperature in the EPC phase [52]. This, in combination with the lowest amount of SF, is limiting the impact strength. Both ELV types are similar in that respect, and the IV(SF) representing the molecular weight of the EPC phase is very similar for all PCRs.The molecular weight of the crystalline fractions is the dominant factor for the MFR of the compositions, but it is also decisive for room-temperature toughness. Only the PCR-ELV-SLFtype shows a higher value for IV(CF), likely as a consequence of the high PE content. This is reflected by the highest Charpy NIS at 23 °C.Mineral filler content is decisive for the stiffness of the compositions, and the PCR-ELV-Bumper type also shows the highest flexural modulus. Secondary factors are EPC content and the PP/PE ratio, as the latter polymer is significantly less stiff.

Taken together, the differences essentially disqualified the PCR-ELV-SLF type for compound development, as no reasonable PCR content could be achieved with the low modulus of this composition. Balancing the EPC resp. overall elastomer and mineral filler content was viable for the other two PCR types, using the experience of normal virgin-based compound development [7,34,53]. The respective virgin polyolefin mixes described in Table 2 are both characterized by rather high SF contents and IV(SF) to improve the toughness and ductility of the resulting compounds, with their slightly different MFRs additionally balancing the overall processability. Variation in the mineral filler content can be seen as a standard measure not only to improve stiffness but also to reduce thermal expansion.

### 3.2. Automotive Compounds

A substantial number of automotive parts for exterior applications such as bumpers are typically composed of different kinds of polypropylenes in combination with external elastomers and fillers. The polypropylenes typically include homopolymers and heterophasic (impact) copolymers, in which the polypropylene matrix provides the stiffness and ethylene propylene elastomer (EPC) ensures the impact performance [34,35]. The external elastomers, mainly low-density ethylene-alpha-olefin copolymers, due to their low glass transition temperature and amorphous nature, contribute to the improvement of the impact behavior at both room and cold temperatures [36,37]. Using the molecular structure and amount of rubber phase, the dimensional stability and various aesthetic properties can be controlled [54]. The role of the inorganic filler is mainly to control the stiffness and thermal expansion, as mentioned above.

The various compound compositions resulted in respective unique phase morphologies controlling the final properties. Figure 4 represents a typical morphology analysis, as carried out by SEM to reveal the heterophasic nature of the blends together with the fine filler distribution. The bright round to slightly elongated objects in the SEM images represent the elastomer phase, composed from a combination of the reactor-based ethylene-propylene copolymer and the external plastomer. The elastomeric phase is well dispersed in the polymer matrix, which appears in black together with fine filler particles in light grey. The structures of all three compounds are similar, but in particular, Compound 2 comprising 40 wt.-% PCR-ELV-Bumper shows more heterogeneity in terms of both elastomer and filler particle size.

To compare the basic mechanical performance, Figure 5a shows the Charpy notched impact strength (NIS) (red bars) and the flexural modulus (blue bars) of the pure different recyclates and the resulting compounds measured at room temperature. Due to the nature of the ELV recyclate, it was possible to incorporate it in a higher amount in relation to the packaging-based material while preserving the stiffness-impact balance of the final material. Compound 2 contains 40 wt.-% of PCR-ELV-Bumper, while Compound 1 can accept just 25 wt.-% of PCR-Packaging in order to achieve a comparable stiffness/impact balance.

For the sake of comparison, the performance of a virgin-based compound, i.e., with 0 wt.-% PCR, is presented to demonstrate the mechanical requirements for the PCR-based compositions. It should be noted that the processability level of said reference in terms of MFR is not reached by any of the PCR compounds, meaning that further development work will be necessary. While less demanding applications are already reachable, this processability issue is still challenging, as increasing the MFR and/or reducing the elastomer-IV of a composition always results in reduced toughness [51,53]. The usual approach for MFR increase for PP is visbreaking, designating a peroxide-induced radical process to reduce the molecular weight, but this process is difficult in case of multiphase systems like impact copolymers and elastomer compounds [35]. As PE itself may branch and crosslink under the influence of radicals, visbreaking such compositions will change the viscosity ratio between the matrix and the elastomer phase, with negative consequences for mechanics and surface quality.

Two further parameters of high relevance for automotive applications, puncture energy measured during the instrumented puncture test performed at 23 °C (blue bars) and the coefficient of linear thermal expansion (CLTE, red bars), are shown in Figure 5b. Compound 2 with the ELV-Bumper recyclate shows, in all cases, puncture energy and CLTE values closer to the virgin grade at higher contents of PCR than Compound 1. Table 3 further shows that the sub-zero impact properties of the recyclate-based compounds, at −20 °C (NIS) and −30 °C (IPT), are also comparable to the virgin reference.

## 4. Summary and Conclusions

The new ELV regulation for the European Union [3] aims at increasing circularity in the automotive industry by imposing mandatory PCR content targets. Mechanically recycled plastics from various waste streams can be used to achieve this target, but at least 25% of the PCR plastic has to originate from end-of-life vehicles. PP plays a major role in this as it is one of the most popular thermoplastics and the main plastic fraction in contemporary cars.

The results, collected in this study, demonstrate that packaging and end-of-life vehicle waste streams are valuable feedstocks to collect PCR PP for automotive applications (see Figure 6). It was found that a compound with a 40 wt.-% ELV-based bumper recyclate can even exceed one with just 25 wt.-% packaging-based recyclate in terms of stiffness/impact balance, which is certainly better than in any of the previous studies [14,28]. The virgin reference can nearly be matched regarding mechanics, but the flowability (MFR) is not reached by any of the PCR compounds, meaning that further development work will be necessary (as explained in detail above, simple visbreaking will not be sufficient).

While the latest generation of mechanically recycled PCR PP from packaging waste is fit to be used in automotive applications already, ELV waste streams need further attention. In particular for the shredder light fraction (SLF), the level of purification needs to be improved before they can enter highly specified automotive applications. The PCR fraction originating from bumper applications seems suitable for closing the loop and being further re-used when it comes to achieve the basic mechanical performance.

## Figures and Tables

**Figure 1 polymers-16-01927-f001:**
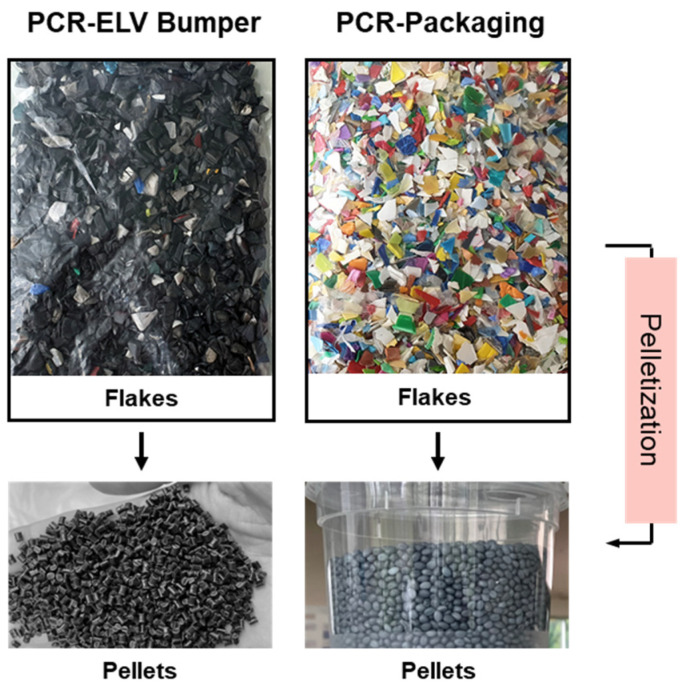
Optical appearance of flakes before compounding and pellets after compounding for PCR-Packaging and PCR-ELV-Bumper materials.

**Figure 2 polymers-16-01927-f002:**
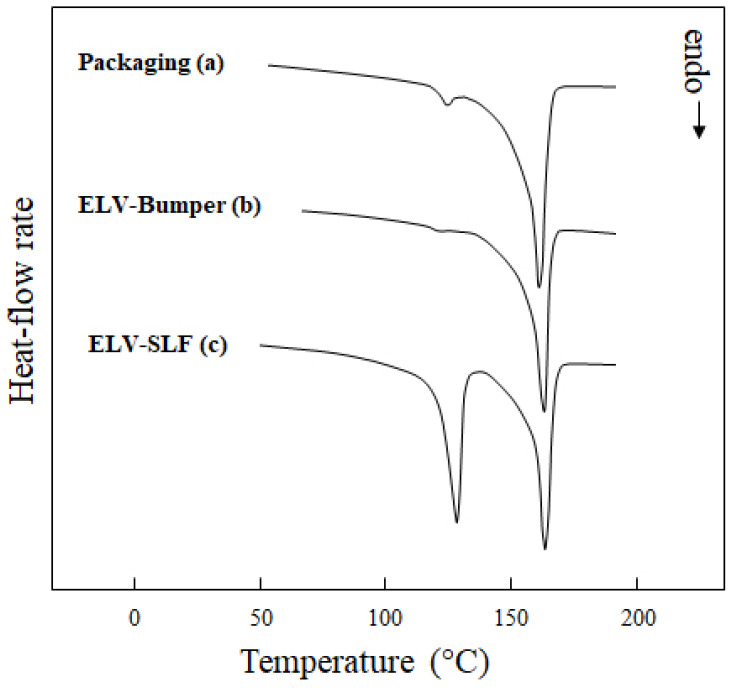
DSC heating scans (2nd heat) of the three PCR materials: (a) PCR-Packaging, (b) PCR-ELV-Bumper, and (c) PCR-ELV-SLF.

**Figure 3 polymers-16-01927-f003:**
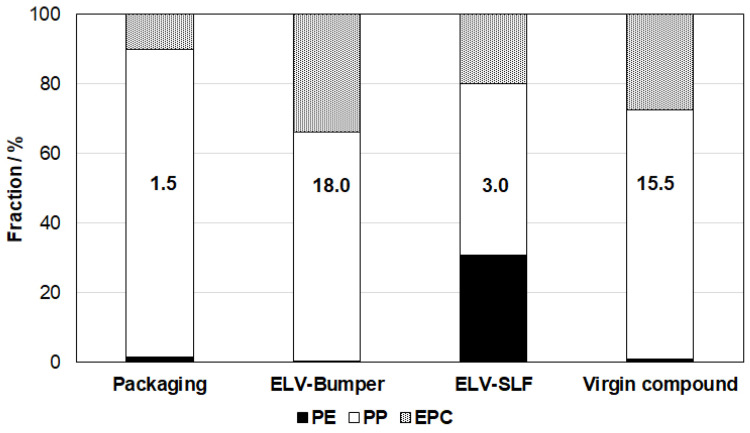
Relative composition of the polymer part of the PCR materials and the reference virgin compound based on DSC and CRYSTEX data (material designations as in Table 1; insert figures indicate ash content).

**Figure 4 polymers-16-01927-f004:**
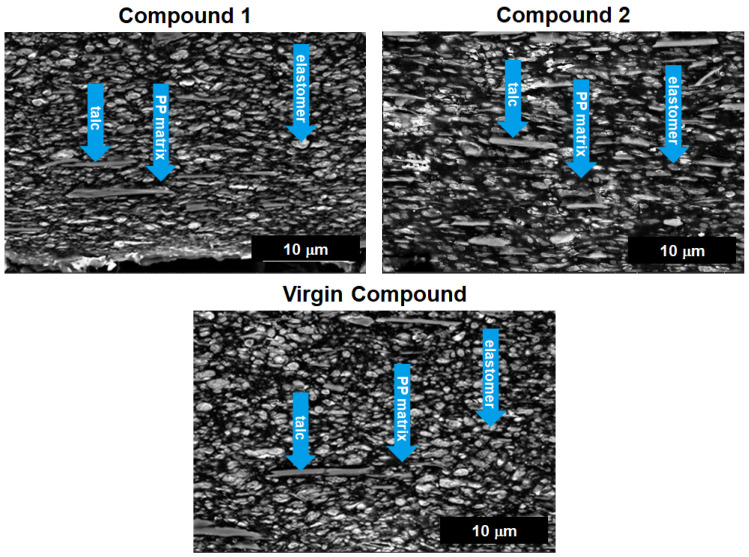
Morphology of tested compounds—SEM images collected from a cross-section of injection-molded specimens in flow direction; inset arrows indicate components.

**Figure 5 polymers-16-01927-f005:**
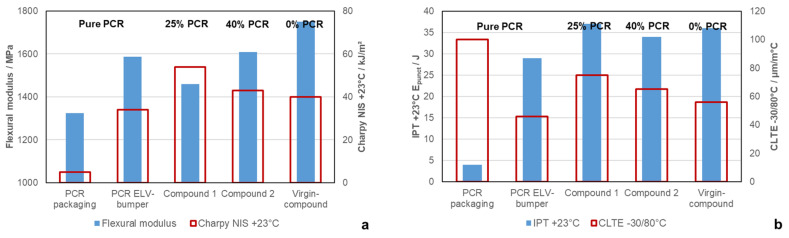
Application performance of the compounded PCRs and resulting compounds in direct comparison to a commercial virgin grade for automotive exterior applications: the (**a**) stiffness/impact balance and (**b**) puncture resistance and coefficient of thermal expansion.

**Figure 6 polymers-16-01927-f006:**
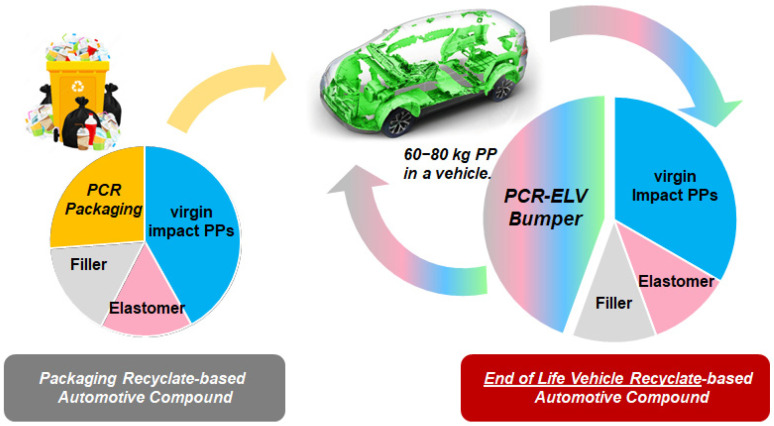
Scxhematic presentation of recyclate stream and contents in the PCR-Packaging based compound 1 and the PCR-ELV-Bumper based compound 2.

**Table 1 polymers-16-01927-t001:** The polymer characteristics of the different recyclates (PCRs).

		PCR-Packaging	PCR-ELV-Bumper	PCR-ELV-SLF
MFR ^1^	g/10 min	25	12	8
Density	kg/m^3^	910	1020	930
Ash ^2^	wt.-%	1.5	18	3.0
DSC				
T_m,PE_	°C	124	124	128
H_m,PE_	J/g	2.5	0.4	56.6
T_m,PP_	°C	163	165	163
H_m,PP_	J/g	95.8	56.6	52.1
CRYSTEX				
SF	wt.-%	10	34	20
C2 total	wt.-%	5.8	23	29
C2(SF)	wt.-%	26	50	41
C2(CF)	wt.-%	3.5	10	27
IV total	dl/g	1.60	1.66	1.95
IV(SF)	dl/g	1.71	1.75	1.74
IV(CF)	dl/g	1.62	1.63	1.96
Mechanics				
Flexural modulus	MPa	1324	1587	637
NIS +23 °C	kJ/m^2^	5.0	34	46
NIS −20 °C	kJ/m^2^	2.2	6.6	8.8
IPT +23 °C E_max_	J	3	18	18
IPT +23 °C E_punct_	J	4	29	29
IPT −30 °C E_max_	J	n.d.	16	13
IPT −30 °C E_punct_	J	n.d.	17	14
CLTE +23/80 °C	µm/m°C	123	55	n.d.
CLTE −30/80 °C	µm/m°C	100	46	n.d.

^1^ at 2.16 kg load, 230 °C; ^2^ TGA residue; n.d.—not determined.

**Table 2 polymers-16-01927-t002:** The polymer characteristics of the virgin polyolefin mixes used for upgrading the recyclates (PCR materials).

		Virgin Polyolefin Mix 1	Virgin Polyolefin Mix 2
MFR ^1^	g/10 min	7.7	9.6
CRYSTEX			
SF	wt.-%	39	30
C2 total	wt.-%	26	12
C2(SF)	wt.-%	55	37
C2(CF)	wt.-%	5.0	5.0
IV total	dL/g	1.8	1.9
IV(SF)	dL/g	2.9	3.53
IV(CF)	dL/g	1.5	1.5

^1^ at 2.16 kg load, 230 °C.

**Table 3 polymers-16-01927-t003:** Conceptual PCR compounds and the resulting properties in comparison to the virgin compound.

		Packaging-PCR Compound 1	ELV-PCR Compound 2	Virgin Compound ^1^
Virgin polyolefin mix 1	wt.-%	58.25		-
Virgin polyolefin mix 2	wt.-%		49.25	-
PCR-Packaging	wt.-%	25.0		-
PCR-ELV-Bumper	wt.-%		40.0	
Talc	wt-%	15.0	9.0	-
Additives	wt.-%	1.75	1.75	-
MFR ^2^	g/10 min	10	9.0	22
Ash ^3^	wt.-%	15.1	17.1	15.0
CRYSTEX				
SF	wt.-%	30	31	-
C2 total	wt.-%	18	16	-
C2 (SF)	wt.-%	50	42	-
C2 (CF)	wt.-%	5.0	7.0	-
IV total	dL/g	1.8	1.9	-
IV (SF)	dL/g	2.6	2.6	-
IV (CF)	dL/g	1.5	1.7	-
Mechanics				
Flexural modulus	MPa	1459	1608	1750
NIS +23 °C	kJ/m^2^	54	43	40
NIS −20 °C	kJ/m^2^	5.9	6	6
IPT +23 °C E_max_	J	20	20	22
IPT +23 °C E_punct_	J	37	34	36
IPT −30 °C E_max_	J	35	17	21
IPT −30 °C E_punct_	J	43	18	31
CLTE +23/80 °C	µm/m°C	90	76	71
CLTE −30/80 °C	µm/m°C	75	65	56

^1^ composition not disclosed; ^2^ at 2.16 kg load, 230 °C; ^3^ TGA residue; n.d.—not determined.

## Data Availability

The original contributions presented in the study are included in the article, further inquiries can be directed to the corresponding author.

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
