# Peer review of "Comparing End-of-Life Vehicle (ELV) and Packaging-Based Recyclates as Components in Polypropylene-Based Compounds for Automotive Applications"

_polymers, 2024, doi:10.3390/polym16131927_

Round 1

Reviewer 1 Report

Comments and Suggestions for Authors

1.     The final paragraph of the introduction should emphasize the novelty of the work.

2.     In this last paragraph, provide a summary of the work, including the methodology and key findings.

3.     Explain the merit and motivation for studying the specific composites.

4.     Have you done any life cycle assessment on the investigated materials?

5.     In the materials section, you mention “PCR, ELV, bumper”. What were the materials/polymers?

6.     What were the used “mineral reinforcement”? Mention the materials in the abstract.

7.     What are the reported parameters in Table 1?

8.     The quality of Figure 1 is not acceptable.

9.     Use arrows to indicate the components in Figure 4.

10. Use bullet points in the conclusion section to highlight the main achievements of the work.

11. Mention the other techniques utilized to fabricate components for automotive and other industrial applications made of recycled materials such as PP. One of those potential methods is 3D printing. Here are some of the recently published works in this area.

Circular economy innovation: A deep investigation on 3D printing of industrial waste polypropylene and carbon fibre composites

3D printing of polypropylene reinforced with hemp fibers: Mechanical, water absorption and morphological properties

Author Response

Please see the respective attchment.

Reviewer 2 Report

Comments and Suggestions for Authors

The manuscript discusses a very relevant topic and is very well written. However, I have some comments and suggestions below for the authors to improve the manuscript below before its acceptance for publication.

Abstract and Title: I am not sure the manuscript title clearly reflects what was investigated. I would suggest the authors to review the title in order to mention that ELV recyclates were compared to a packaging-based PCR for automotive applications.

Lines 22-38: There was a lot of information regarding the European context, but it was not specified in the title that the focus would be in Europe. Additionally, the journal audience is global. I would strongly suggest the authors to include information that represents other regions or global numbers (e.g., how much PP was produced globally? Or does other countries/regions also have targets for recycled plastic content in cars?)

Line 139: What is a “(semi)commercial” recyclate? It is commercially available or not. 

Lines 146-148: Is the light fraction PCR-ELV from the Netherlands also originated from bumpers? I did not understand the light fraction terminology. The density value showed in Table 1 is similar to common unfilled polypropylene.

Lines 149-150: I have never seen the use of the term “double-screw extruder”. Is that a twin-screw extruder? Also, please include extruder diameter, L/D, pelletization mode, temperature profile and screw speed for reproducibility. Was the temperature profile the same for all three PCR resins evaluated? What kind of melt filter was used? Please include model/manufacturer information.

Lines 168-169: What is the composition of the virgin polyolefin mix 1 and 2 used in the study? 

Lines 212-222: Information regarding the equipment used for injection molding the specimens and for the mechanical tests is missing. Please include model/manufacturer. 

Lines 224-226: Include SEM instrument information. 

Lines 229-231: Please include L/D and screw speed of the extruder. 

Lines 331-335: The MFR of the virgin compounds used to mix with the PCR were lower than the virgin resin used as a control. Why a virgin compound with higher MFR was not used if processability (MFR) is necessary to match the MFR = 22 g/10 min of the virgin resin (0% PCR)?

Lines 347-350: These targets are for Europe. Please mention this fact.

Lines 361-363: What are the authors suggestions to improve flowability (MFR) of the PCR compouds? For example, could peroxide visbreaking be an alternative? Or improve PCR sorting to obtain a PCR fraction with higher MFR (possibly containing high-MFR thin-walled PP containers)?

Author Response

Please see the respective attachment.
